# Backyard Poultry Flocks in Morocco: Demographic Characteristics, Husbandry Practices, and Disease and Biosecurity Management

**DOI:** 10.3390/ani13020202

**Published:** 2023-01-05

**Authors:** Asma Fagrach, Siham Fellahi, Mohammed Kamal Challioui, Oumaima Arbani, Ibtissam El Zirani, Faouzi Kichou, Mohammed Bouslikhane

**Affiliations:** 1Department of Pathology and Veterinary Public Health, Institut Agronomique et Vétérinaire Hassan II, Rabat P.O. Box 6202, Morocco; 2Animal Production Department, Institut Agronomique et Vétérinaire Hassan II, Rabat P.O. Box 6202, Morocco

**Keywords:** backyard poultry, flock owners, disease risk factors, disease management, biosecurity

## Abstract

**Simple Summary:**

No previous studies have focused on describing the current situation of backyard poultry flocks in Morocco and its potential risks to the commercial sector and public health. The results of this survey revealed that backyard poultry farming is proving itself to play a major role in the maintenance and spread of diseases due to the lack of vaccination, lack of veterinary consulting, lack of biosecurity practices (such as poor hygienic conditions), and irrational self-medication of diseased birds using antibiotics, pesticides, and hazardous chemicals that could be a significant health risk for consumers. To mitigate the risks of backyard poultry flocks on the commercial sector and public health, outreach programs about disease prevention and biosecurity practices, along with prophylactic campaigns, should be implemented.

**Abstract:**

Backyard poultry farming is an important tool for poverty alleviation and food security in rural areas of Morocco. A descriptive epidemiologic survey was conducted in 286 backyard poultry flocks from the provinces of Khemisset and Skhirat-Temara to gain baseline data on the current status of backyard poultry flocks in Morocco as well as its potential implications on the transmission and spread of avian diseases. The findings indicated that 88.8% of flocks were raised in a mixed confinement system, with an average flock size of 30 birds (range 1–352). Chickens accounted for 83% of the overall reported birds. More than two-thirds of respondents (69%) kept chickens only, while the remaining flocks raising multiple bird species in total promiscuity. Diseases were the highest cause of mortality (84.7%), followed by predation (15.3%). According to 56.1% of the owners, respiratory symptoms were among the major disease signs reported, besides ectoparasite infestation. Flock health management revealed a lack of preventive vaccination, lack of veterinary consulting, lack of biosecurity practices, and irrational self-medication of diseased birds using antibiotics, pesticides, and hazardous chemicals that could be a significant health risk for consumers. The need for an outreach program about disease prevention and biosecurity practices, along with prophylactic campaigns, should be emphasized to further mitigate the risks of backyard poultry flocks on the commercial sector and public health.

## 1. Introduction

In Morocco, poultry production has experienced a great growth over the past four decades; production increased from 55,000 tons and 278 million eggs in 1981 to 625,000 tons and 5.5 billion eggs in 2021 [1]. The poultry industry is dominated by commercial farms, while the contribution of backyard poultry farming to total national production remains minor. According to the Interprofessional Federation of the Poultry Sector, 50,000 tons of white meat and 800 million eggs would come from backyard poultry farming in 2021. However, it still occupies an important and promising position, as it may contribute to the achievement of several of the Millennium Development Goals, particularly the improvement of food security, income generation, women’s empowerment, and poverty alleviation [2], and to the capacity of the rural population to deal with different crisis situations that have become recurrent, such as poor harvests, drought, and climatic disasters. Backyard poultry farming allows vulnerable families to ensure a constant source of animal proteins and to generate income through the marketing of birds and eggs. Moreover, consumers’ general preference for products of local chickens kept in free-range conditions has contributed to the sustainability of this activity [3]. Despite its valuable role, little is known about this activity and its potential implications on the transmission and spread of avian and zoonotic diseases, which are a public health concern. The role of domestic birds in the transmission of diseases is not well investigated. However, it is believed that backyard poultry can lead to the introduction and spread of infectious diseases such as Avian influenza and Newcastle disease, since backyard poultry farming represents the interface where interaction between domestic and wild birds occurs [4,5]. Moreover, these diseases can be zoonotic, with fatal consequences in both poultry and humans, or can represent a serious risk to commercial poultry systems and international trade [6]. Furthermore, backyard poultry owners’ lack of experience and knowledge about biosecurity and hygiene practices may bring additional risk concerning other zoonotic and food-borne pathogens, such as Salmonella and Campylobacter [7,8]. Hence, knowledge about the characteristics and management practices, including biosecurity measures and disease management, in backyard poultry flocks is therefore of high value for national veterinary authorities in charge of disease control programs and contingency plans in case of emergent diseases, as well as for decision making in policy development. Therefore, this study aims to gain insights into the characteristics and management practices of backyard poultry flocks in two areas of Morocco (Khemisset and Skhirat-Temara provinces), with a special focus on biosecurity practices and disease management.

## 2. Materials and Methods

### 2.1. Study Design

The surveys were conducted in two locations: Khemisset and Skhirat-Temara. The two locations are predominantly rural areas that surround Rabat, the capital city, and so the demand for backyard poultry products is much higher than in other provinces, and to meet urban consumer requirements—and to take benefit from the strongly growing demand—almost every rural household has a small poultry flock in their backyard. For this study, multistage sampling technique was used to select the respondents. Firstly, county selection was based on the abundance of backyard flocks as well as density of commercial poultry farms. Secondly, random sampling technique was used to select the respondents. Two hundred and eighty-six backyard poultry flock owners were randomly selected. The number of respondents sampled per rural county was proportional to the number of households registered in each locality according to the last census [9].

### 2.2. Data Collection and Analysis

Data were collected from 286 backyard flocks located in 13 rural counties, namely Brachoua, Moulay Driss Aghbal, Jemaat Moul Blad, Marchouch, Aîn Sbit, Maâziz Ezzhiliga, and Had Lagwalem in the province of Khemisset, and Ain El Aouda, Oumazza, El Menzeh, Sidi Yahia Zaer, Mers El Khir, Ain Atiq, and Sabbah in the province of Skhirat-Temara (Figure 1). A structured questionnaire was used to gather field information through face-to-face interviews. A pilot field survey was performed to evaluate the questionnaire. Accordingly, revisions and adjustments were made. Participation in the survey was voluntary, and before each interview, the investigator explained to the flock owner the purpose of the study, its context, and expectations, and obtained their informed consent. Participants were asked to report information on the size and structure of poultry flocks, purposes for raising chickens, housing, and feeding system, as well as health problems and biosecurity measures. Excel^®^ software (2016 version) was used for data gathering and basic analysis. The Khi2 test in IBM SPSS Statistics 26 (SPSS Inc., Chicago, IL, USA) was conducted to assess for significant differences between the calculated frequencies and proportions. Then, Khi2 test of independence was performed to verify possible associations between variables. The significance threshold for all statistical tests was set at 5%.

## 3. Results

### 3.1. Respondents’ Demographics

The respondents are assumed to be the ones in charge of poultry raising. Women made up 86% of the flock owners. Their ages vary between 16 and 84 years, with an average of 45.2 years, and the majority (77.6%) of them are married. About 72% of poultry farmers are illiterate, while 16.4% have a primary education, 11.2% have a secondary school level, and only 0.7% of poultry farmers have a higher degree. Agriculture and livestock are the most important subsistence activities. Over 75% of flock owners have been raising poultry for more than 10 years (Table 1).

### 3.2. Backyard Flocks’ Characteristics

From the survey’s data, a total of 8452 birds were recorded on all flocks investigated, including 7019 (83%) chickens, 609 (7.2%) guinea fowls, 526 (6.2%) pigeons, 216 (2.6%) turkeys, and 82 (0.9%) ornamental birds comprising peacocks, geese, and ducks. Flock sizes ranged from 1 to 352 birds, with a average of 30 birds per flock (Table 2). The average chicken flock size was 25, with a wide range of 1 to 200. All owners reported having chickens, with more than two-thirds of farmers (69%) keeping chickens only. The remaining backyard flocks had multiple bird species raised together in total promiscuity. Nearly 87% of the households raised poultry for self-consumption and income generation, in 10.8% of flocks, it was reported that poultry was not subject to any sale, and in only 2.5% of farms were the products intended exclusively for sale. Almost 87.4% of poultry owners reported that they purchase new birds from rural markets, while 12.6% reported that home-hatched was the only source of birds (Table 2).

### 3.3. Flock Management

In 88.8% of the visited farms, mixed confinement was the prevailing system, with birds roaming free during the day and being confined at night. Night shelters made with equipment usually found on site are very basic, poorly designed, very airy, and leaky to wild birds and rodents. This situation is exposing birds more to predation and bad weather. Indeed, more than 15% of flock owners reported bird losses due to predators, mainly raptors and rats. These shelters were poorly maintained, not disinfected, and rarely cleaned in 63.4% of the surveyed flocks, while in 36.6% of them, cleaning was practiced frequently.

Backyard birds are fed basically on what they find in their environment (seeds, insects, worms) from scavenging. However, 96% of flock owners provided supplementation to their birds. The most common supplements are barley and stale bread (93.6% and 81.3%, respectively), followed by leftover kitchen waste, wheat bran, and wheat, with 65.4%, 62.9%, and 45.2%, respectively. The least common were commercial compound feed (20.1%) and corn (19.8%) (Figure 2). More than 98% of the farmers surveyed did not use a feeder and instead fed their chickens by throwing the feed on the ground or the surface of recycled materials such as plastic sheets. Regarding watering, 94.7% of owners provided ad libitum drinking water to their birds. The most common source of drinking water for the birds was wells and/or public drinking water networks, while in 5.3% of flocks, birds obtained water directly from environmental sources. Among flock owners providing drinking water to their birds, 97.1% used recycled material containers and 2.9% used commercial drinkers. The quality of the water provided was poor, due to dirty drinkers that were rarely cleaned or unsafe water sources. About 58.6% of the surveyed farmers said that they do not clean their feeders and drinkers.

### 3.4. Flock Health

According to the farmers, diseases are the main cause of mortality (97.6%), followed by predation (15%) generally caused by raptors and rats. None of the farmers had received formal training in disease recognition. All owners also reported observing signs of disease in their flock within the last 6 months. Almost 98.3% of owners reported seeing birds with respiratory symptoms such as rales, sneezing, dyspnea, and cyanosis, especially in the cold season, whereas 39.5% reported seeing the appearance of nodules on the head and legs of birds, and 39.1% observed birds with greenish diarrhea. Eighty-seven owners (30.4%) reported seeing locomotor disorders such as lameness. Prostration and anorexia were reported in 28.3% of flocks, while neurologic signs such as torticollis and paralysis were reported in 4.2% of flocks.

### 3.5. Vaccination and Medication

Flock owners’ responses to vaccination and medication are shown in Table 3. It appears clearly that most farmers (99.3%) had never vaccinated their poultry flocks. Control of ectoparasites was reported in 88.8% of the surveyed flocks. The pesticides used are generally very powerful phytosanitary products applied directly to birds. Regarding disease management, 88.1% of flocks did not receive any veterinary care. For the treatment of diseased birds, 43.4% of the owners declared giving drugs such as antibiotics and vitamins to their sick birds without any veterinary supervision; the antibiotics used are generally intended for other livestock species, or human use. Almost 28% of flock owners treated sick birds with ethnomedical remedies such as onions, nettles, and spices, and in 28.7% of flocks, no treatment of sick birds was performed. Results from the Khi2 analysis indicated that the practice of vaccination is related to education level (*p* < 0.05) and flock owner gender (*p* < 0.001), and the lack of veterinary consultation is related to the presence of industrial poultry farms nearby (*p* < 0.001).

### 3.6. Biosecurity Conditions

The most common system of rearing poultry was in a mixed housing system (88.8%), which consisted of housing the birds overnight and letting them be free-range during the day. That is to say that in 88.8% of the visited flocks, poultry could move freely, and in 76.5% of the systems, contact with wild birds such as sparrows was observed. Results from the Khi2 analysis (*p* < 0.05) indicated that contact with wild birds is related to the confinement system and water source adopted by the flock owners. In addition, it was possible to identify that in all the flocks surveyed (100%), visitors could get in close contact with the poultry raised, and no disinfection procedure was applied before entering or leaving a farm. In addition, 38.6% of the farms were located near commercial poultry farms. To manage on-farm mortalities, the most common disposal method was throwing dead birds off-farm (79.7%), followed by the burning or burial of dead birds (20.3%). When new birds are introduced to a farm, 69.2% of farmers say they do not initiate any quarantine procedures and they have no concern nor awareness about the health of the new birds or the health status of the farms where they come from (Table 3).

## 4. Discussion

Backyard poultry production is an omnipresent activity among rural households in the developing world. It is considered an effective way to ensure food security, improve farmers’ income, and promote women’s empowerment [10]. However, backyard poultry flocks have long been presumed to play a role in the transmission and spread pathways of avian diseases. To our knowledge, there has been little documented information on backyard poultry flocks in Morocco, and significantly less emphasis has been placed on assessing disease risk and biosecurity conditions. The study aimed to describe the characteristics and management practices of backyard poultry flocks in Morocco.

The survey respondents were mainly women. This is consistent with previous findings in other studies in African countries including Algeria [11,12], Mauritania [13], and Egypt [14], where it is usual for poultry to be owned and managed by women. This could be explained by the fact that poultry keeping is neglected by men because it is mainly oriented toward home consumption and does not make a great profit. A clear lack of education among poultry keepers was revealed throughout the survey, as in most developing countries [15,16]. This lack should be constantly considered in any project targeting the improvement of poultry-keeping practices and organizing producers [17].

The main purposes for keeping birds were the home consumption of eggs and meat and the generation of income, suggesting some consistency between developing countries in terms of motivation for keeping poultry [18,19,20,21], where the main reasons are to generate income and as a food source, which ensure the greater role of backyard poultry in food security and poverty alleviation of rural populations [22].

Furthermore, 31% of respondents stated that they also raised other poultry species such as guinea fowl and pigeons. In accordance with other studies, it is usual for backyard poultry keepers to maintain bird species besides chickens [20,23,24,25,26]. This is relevant, considering that multiple bird species raised together in total promiscuity can be associated with a higher risk of diseases such as Newcastle disease and avian influenza [27,28,29].

Moreover, it was observed that almost 70.9% of the respondents kept other livestock species alongside poultry, most commonly cattle, sheep, and equine. Mixed farming is obviously increasing the risk of disease transmission between species. The higher risk would be West Nile disease transmission between birds and equines, especially when 40.1% of the respondents reported having equines alongside poultry [30].

Housing is important for protecting poultry from bad weather and wild animals. It was reported in this study that birds were kept free-range during the day and in shelters at night. However, these chicken shelters were in poor hygienic conditions and did not provide adequate protection from cold and predators at night. This mixed confinement system creates conditions that increase the opportunity for contact between poultry of different flocks, and as shown in the results, increase their interactions with wild birds, and consequently, the risk of contamination and spread of diseases. Previous studies have already suggested that contact at the wild–domestic bird interface is more likely in free-range farming systems [19,31].

The interviewed owners do not practice any rational feeding system. Birds feed on what they find during scavenging. The supplement, consisting of barley grains and stale bread, is generally provided in insufficient quantities. Moreover, it lacks any vitamin or mineral supplies. It has already been reported that in developing countries, chickens are kept with very low inputs in the villages. Besides scavenging, supplements such as crushed grains, raw or boiled rice, and leftover kitchen waste are also offered to the birds [32]. Almost 5.3% of owners do not provide drinking water to their poultry, and birds obtain water from existing environmental sources such as streams and ponds, which could increase the likelihood of contact with wild birds. Moreover, these water sources may have been contaminated by infected carcasses that were thrown into the water or by feces from passing wild birds, which could be a potential indirect pathway for pathogen transmission between wild birds and domestic poultry. Avian influenza viruses can survive in water for long periods of time [33]. Such a transmission pattern was implicated in an outbreak of highly pathogenic avian influenza affecting commercial poultry in Australia [34].

As a result, diseases were the most common cause of death, and the main health problem reported in the past six months was external parasites, indicating why treatment of external parasites was the most reported health practice. Moreover, 98.3% of owners also reported observing respiratory signs in their flock within the last 6 months. This high disease level is probably due to the exposure of birds to the natural environment, the interaction of different bird species within and among flock contacts during scavenging, the uncontrolled introduction of new birds, contacts through the exchange or sale of live chickens, and/or movement between households and villages. Interestingly, gender and education level seem to be related to vaccination adoption. Vaccination in this study was mainly made by men and by flock owners with high education level. Gender and education level were found to be the most important variables affecting the adoption decision of management interventions in indigenous chicken production in Kenya [35]. Another finding related to disease management was that despite the widespread health problems, only 11.9% sought veterinary care. The main reasons for not seeking veterinary care were the cost, the lack of veterinarians with backyard poultry experience, and that flock owners—especially those who lived near commercial poultry sets—thought that they could manage the problem on their own based on the knowledge on avian diseases and treatment methods that they acquired through their work in these farms or through the people who worked there. That is why sick birds were generally treated without any veterinary supervision, using antibiotics and other drugs approved for other species or for human use, which could represent a high risk to human health, owing to the possible presence of drug residues in poultry products. Kichou et al. (2000) [36] also reported the use of contraceptive pills to treat swollen heads and diarrhea and to stimulate growth in sick chicks. The use of such inappropriate treatments may be attributed to illiteracy, poverty, a lack of knowledge of basic health and management practices, and a lack of institutional interventions and monitoring. Furthermore, ethnoveterinary medicine is still practiced because of its affordability and availability for resource-poor farmers [37].

Regarding biosecurity measures, it was recognized in the present work that 79.7% of the respondents indicate that they dispose of dead birds’ carcasses outside the farm, even in water streams, which is a practice that would obviously represent a source of infection for backyard poultry, wild birds, and commercial flocks [38,39]. In the present study, it is optimistic that 61.4% of the backyard poultry flocks were not close to commercial poultry farms. This might decrease—but not completely eliminate—the risk of disease spread from a backyard to a commercial setting.

Furthermore, while weekly local markets offer a great opportunity for poultry owners to sell their products, they also constitute a serious health concern, because chickens could easily get infected or infect other birds. Purchasing new birds or returning unsold chickens from local live poultry markets without applying quarantine procedures, as mentioned by a large part of respondents (69.2%), could be another obvious risk factor for disease introduction and transmission. Altogether, the lack of biosecurity measures observed in this study, such as mixed confinement, lack of veterinary assistance, and incorrect handling of dead birds, among others, are consistent with other international reports, especially from developing countries where such conditions are common [23,29,40].

## 5. Study Limitations

The study has some limitations. As in any observational study, this study is not free from potential bias using surveys and interviews, even if some health management and biosecurity practices were audited. Additionally, a larger sample size might have improved the study’s capacity to identify meaningful associations between variables.

## 6. Conclusions

Backyard poultry flock owners were found to be uninformed about health management and biosecurity practices; this might result in exposing this poultry system to a high risk of introduction and spread of infectious diseases that could pose a public health concern, such as highly pathogenic avian influenza. The findings of this study highlight the need for an extension program that develops poultry owners’ knowledge and skills about good husbandry practices, health management, and biosecurity measures. Furthermore, this work is an alert message to the livestock and veterinary authorities to improve coverage of veterinary assistance and surveillance activities in backyard poultry, and emphasizes the need for effective policies that are specifically addressed to backyard poultry farming that consider the risks to commercial poultry flocks and public health and the importance of this activity to the food security and wellbeing of rural populations.

## Figures and Tables

**Figure 1 animals-13-00202-f001:**
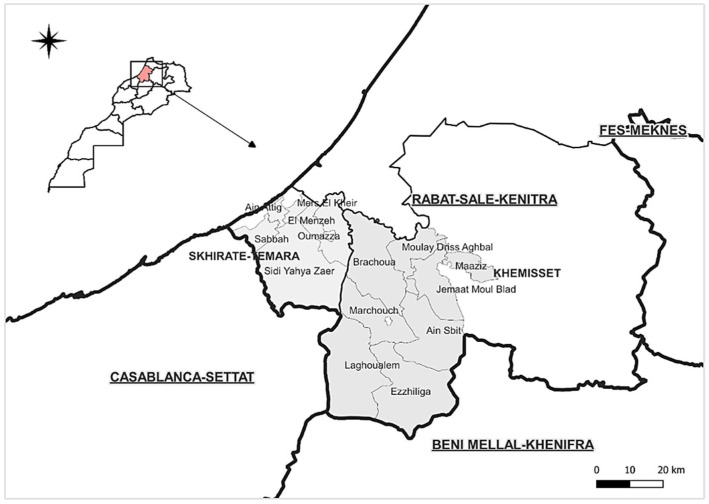
Map of surveyed counties (indicated by grey shades).

**Figure 2 animals-13-00202-f002:**
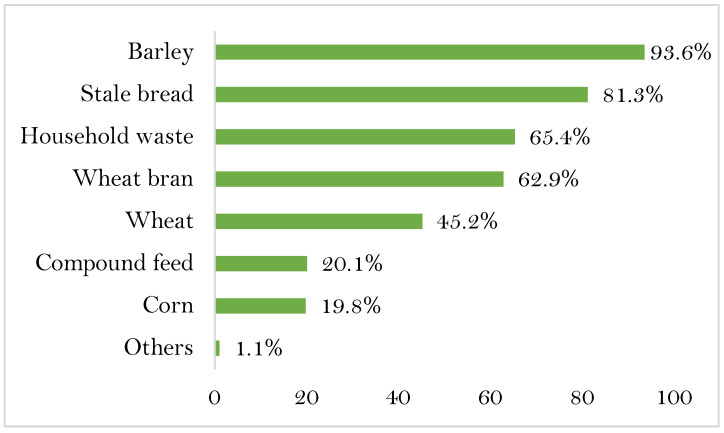
Main feed used for chicken feeding in percentage.

**Table 1 animals-13-00202-t001:** Percentage of backyard flock characteristics as reported by flock owners.

Characteristic	%	*p*-Value
**Location**		<0.05
Khemisset	55.9	
Skhirat-Temara	44.1	
**Flock owner**		<0.001
Women	86.4	
Men	13.6	
**Level of education**		<0.001
Illiterate	71.7	
Primary	16.4	
Secondary	11.2	
University	0.7	
**Experience in raising poultry**		<0.001
≤10 years	75	
>10 years	25	
**Poultry farming training received**		<0.001
No	99.3	
Yes	0.7	
**Reason for poultry farming**		<0.001
Self-consumption and income generation	86.7	
Self-consumption	10.9	
Income generation	2.5	
**Confinement system**		<0.001
Mixed	88.8	
Alternate	11.2	
**Source of birds**		<0.001
Home-hatching	12.6	
Rural markets	87.4	
**Flocks with only chickens**	69	<0.001
**Flocks with multiple species**	31	
**Cleaning and disinfection of shelters**		<0.001
Yes	63.4	
No	36.6	
**Cleaning frequency**		<0.001
Once a week	79.3	
Once a month	20.7	
**Food supplementation**		<0.001
Yes	96	
No	4	
**Frequency of supplementation**		<0.001
Twice a day	62.9	
Once a day	17.3	
Three times a day	18.4	
**Method of distribution**		<0.001
Discarded on the ground or salvage material	98.6	
Commercial feeders	1.4	
**Drinking water sources**		<0.001
Wells and/or public drinking water network	94.7	
Environmental sources	5.3	
**Cleaning of feeders and/or drinkers**		>0.05
Yes	41.4	
No	58.6	
**Cleaning frequency**		<0.001
Once a day	87.9	
Once a week	12.1	
**Causes of mortality (last year)**		<0.001
Predation	15.3	
Disease	84.7	
**Signs of disease (last 6 months)**		
Parasitism	100	
Respiratory (rales, sneezing, dyspnea, cyanosis)	98.3	
Nodules on head and legs	39.5	
Greenish diarrhea	39.1	
Locomotor disorders (lameness)	30.4	
Prostration and anorexia	28.3	
Neurologic (torticollis, paralysis)	4.2	

**Table 2 animals-13-00202-t002:** Backyard flock species distribution and average flock size among the surveyed area.

	Khemisset	Skhirat-Temara	Overall (%)	Range
**No. of flocks**	160	126	286	
**Average flock size**	17	20	29.7 ± 4.4	1–352
**Average chicken flock size**	16	20	24.7 ± 3.3	1–200
**Species**	4644	3808	8452	-
Chickens	3632	3387	7019 (83%)	1–200
Guinea fowl	399	210	609 (7.2%)	0–100
Pigeons	385	141	526 (6.2%)	0–80
Turkeys	168	48	216 (2.6%)	0–20
Geese	36	8	44 (0.5%)	0–14
Ducks	16	12	28 (0.3%)	0–12
Peacocks	8	2	10 (0.1%)	0–8

**Table 3 animals-13-00202-t003:** Flock owners’ responses to poultry health management and biosecurity practices in percentage.

Practices	%	*p*-Value
**Vaccination**		<0.001
No	99.3	
Yes	0.7	
**Ectoparasite treatment**		<0.001
Yes	88.8	
No	11.2	
**Veterinary care**		<0.001
Yes	11.9	
No	88.1	
**Treatment of sick birds**		<0.001
Drugs (antibiotics, vitamins)	43.4	
Ethnomedical remedies	27.9	
No treatment	28.7	
**Disposal of dead birds**		<0.001
Thrown	79.7	
Destroyed	20.3	
**Quarantine application to the introduction of new birds**		<0.001
No	69.2	
Yes	30.8	
**Contact with wild birds**		<0.001
Yes	76.5	
No	23.5	
**Contact with livestock**		
Cattle	52.5	
Sheep	47.5	
Goats	17.3	
Equines	40.1	
Rabbits	2.8	
**Presence of industrial poultry farms nearby**		<0.001
Yes	38.6	
No	61.4	

## Data Availability

The datasets used and/or analyzed during the current study are available from the corresponding author on reasonable request.

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
