# Peer review of "Backyard Poultry Flocks in Morocco: Demographic Characteristics, Husbandry Practices, and Disease and Biosecurity Management"

_animals, 2023, doi:10.3390/ani13020202_

Round 1

Reviewer 1 Report

Dear authors,

The manuscript title is “Backyard poultry flocks in Morocco: Demographic characteristics, husbandry practices, and disease and biosecurity management” and it intends to characterize the management and biosecurity practices of backyard poultry flocks in two regions of Morocco.

The topic falls within the aims and scope of the journal and I do believe we need to look better to backyard production, so this is a nice topic.

Some particular suggestions/comments will be done here:

-      Line 13 – delete study or survey

-      Line 14 – delete preventive

-      Line 15 – lack of biosecurity practices include the poor hygienic conditions so maybe you can delete poor hygienic conditions or write “lack of biosecurity practices (like poor hygienic conditions)”

-      Lines 18 and 19 – the same as before: disease prevention, biosecurity measures and prophylactic campaigns are almost synonyms, so you can summarize

-      Lines 23 to 28 – please be consistent in the number of decimal cases of the percentages (in all manuscript); please write “.” Instead of “,”, for example: 84.7% instead of 84,7% (in all manuscript)

-      Lines 28 – 35 – the same as told before in lines 13 – 19

-      Line 36 – please do not repeat words that are in the title in keywords

-      Line 44 – first time reported, please write FISA in full

-      Lines 64 – 65 – please write in italic Salmonella/Campylobacter

-      Line 81 – what do you mean with “relative concentration of commercial industries”? What kind of industries? And why was this used to define the sampling methodology?

-      Line 84 – 85 – I also have doubts about this “random” sampling and the number of respondents being proportional to the households because maybe where there are more households there are less farmers (more households, more urban?). Also I do not understand how did you locate the farmers? Because not all the households have backyard flocks for sure. Did you contact any animal association? Veterinary official department? How did you find the farmers?

-      Line 87 – You have 286 flocks or farmers? Any of theses farmers had more than one flock?

-      Table 1 – what do you mean with housing system mixed and alternate? And with food supplementation? About the cleaning questions, more than yes/no you need to know the frequency. Because this way it tells us nothing. I mean, for example, if a farmer cleans once in decade, that is nothing. Endoparasite is infection and not infestation.

-      Line 118 – all ducks are ornamental in Morocco?

-      Line 136 – what kind of cleaning? Because if the floor is on land/soil maybe the cleaning is just a remove of the faecal matter, no?

-      Line 141 – something is missing in the end of this line or not?

-      Lines 145/6 – ad libitum (in italics)

-      Lines 152,157, 167 – delete reference to table 1 here

-      Line 154 – legend: add “in percentage”

-      Lines 155 – 167 – If you have all this information in the Table as it seems, you should delete this; you should not duplicate information in text and tables

-      Line 176 – please do not begin a phrase with a number

-      Line 177 – can you add some examples of what you called ethnomedical remedies?

-      Line 180 – please add “in percentage”

-      Table 3 – what kind of vaccination was mentioned? I believe it is important to add; what do you mean with “thrown”? Cattle instead of cattles

-      Lines 181 – 193 – the same as in lines 155 – 167

-      Discussion: I arrived the discussion waiting to read some statistical significance tests of association between your variables; I believe you need to improve your study with some more statistical content than percentages, there are so many variables to associate!

-      Line 246 – AI in full (or in first report write it full plus (AI))

-      You should add the limitations of your study and I believe you can enrich the discussion much more with a better statistical analysis.

-      Line 408 – delete 39

Author Response

Response to Reviewer 1 Comments

We thank the Editor and Reviewers for their valuable comments and the opportunity to improve our manuscript and resubmit to Animals. We have carefully considered the comments and revised the manuscript accordingly. The following is a point-by-point response to the all the Editor and Reviewers’ questions and comments.

Answer to Editorial comments

Make changes as recommended in the reviewer's critiques

Reviewer #1:

Line 13 – delete study or survey

The word “study” is now removed.

Line 14 – delete preventive

The word “preventive” is now removed.

Line 15 – lack of biosecurity practices includes the poor hygienic conditions so maybe you can delete poor hygienic conditions or write “lack of biosecurity practices (like poor hygienic conditions)”

The Reviewer is correct.

Lines 18 and 19 – the same as before: disease prevention, biosecurity measures and prophylactic campaigns are almost synonyms, so you can summarize

We thank reviewer for this comment. However, in this sentence we mean that the flock owners need an outreach program about disease prevention and biosecurity practices and the veterinary services should implement prophylactic campaigns

Lines 23 to 28 – please be consistent in the number of decimal cases of the percentages (in all manuscript); please write “.” Instead of “,”, for example: 84.7% instead of 84,7% (in all manuscript)

The Reviewer is correct.

Lines 28 – 35 – the same as told before in lines 13 – 19

The sentence “poor hygienic conditions” is now removed.

Line 36 – please do not repeat words that are in the title in keywords

The Reviewer is correct.

Line 44 – first time reported, please write FISA in full

The Reviewer is correct. The Word FISA in now changed with the english name of the organization

Lines 64 – 65 – please write in italic Salmonella/Campylobacter

Done

-      Line 81 – what do you mean with “relative concentration of commercial industries”? What kind of industries? And why was this used to define the sampling methodology?

The Reviewer is correct. This sentence in now changed to “high density of commercial poultry farms”

-      Line 84 – 85 – I also have doubts about this “random” sampling and the number of respondents being proportional to the households because maybe where there are more households there are less farmers (more households, more urban?). Also, I do not understand how did you locate the farmers? Because not all the households have backyard flocks for sure. Did you contact any animal association? Veterinary official department? How did you find the farmers?

We thank reviewer for this comment. In the rural areas of Morocco, every household had a small poultry flock in their backyard, that it why it was not difficult to find households that have a backyard poultry, we did not contact any association or veterinary services, but we were accompanied with the county chief to introduce us to the respondents.

-      Line 87 – You have 286 flocks or farmers? Any of these farmers had more than one flock?

We have visited 286 households; every household (farm) had a small poultry flock in their backyard. The respondent was the person in charge of raising poultry

-      Table 1 – what do you mean with housing system mixed and alternate? And with food supplementation? About the cleaning questions, more than yes/no you need to know the frequency. Because this way it tells us nothing. I mean, for example, if a farmer cleans once in decade, that is nothing. Endoparasite is infection and not infestation.

Mixing Housing system means that the poultry was kept free during the day and sheltered at night

The alternate housing system was that poultry was kept sheltered during cold seasons and free during the hot seasons.

The supplementation means that besides the scavenging, the flocks owners provide food supplementation mainly composed of barely and stale bread.

The reviewer is correct we have added the frequency of cleaning the shelters and the feeders and drinkers

-      Line 118 – all ducks are ornamental in Morocco?

We thank the reviewer for this comment. From our meeting with the respondents, we have decided to classify the ducks with the ornamental birds because the person we have interviewed kept them for ornamental purposes and not for consumption

-      Line 136 – what kind of cleaning? Because if the floor is on land/soil maybe the cleaning is just a remove of the fecal matter, no?

The cleaning means the remove of fecal dejections

-      Line 141 – something is missing in the end of this line or not?

I think noting is missed the sentence is “The most common supplements are barley and stale bread (93,6% and 81,3%, respectively), followed by leftover kitchen waste, wheat bran, and wheat with 65,4%, 62,9%, and 45,2% respectively.”

-      Lines 145/6 – ad libitum (in italics)

Done

-      Lines 152,157, 167 – delete reference to table 1 here

Done

-      Line 154 – legend: add “in percentage”

Done

-      Lines 155 – 167 – If you have all this information in the Table as it seems, you should delete this; you should not duplicate information in text and tables

We thank the reviewer for this comment. However, I found that this paragraph is important because it gave more details and clarifications to the table content.

-      Line 176 – please do not begin a phrase with a number

The sentence is corrected

-      Line 177 – can you add some examples of what you called ethnomedical remedies?

Done

-      Line 180 – please add “in percentage”

Done

-      Table 3 – what kind of vaccination was mentioned? I believe it is important to add; what do you mean with “thrown”? Cattle instead of cattles

During the survey, some of the respondents vaccinate their poultry against multiple diseases.

Thrown means that the dead birds are thrown off the farm

-      Lines 181 – 193 – the same as in lines 155 – 167

We thank the reviewer for this comment. However, I found that this paragraph is important because it gave more details and clarifications to the table content.

-      Discussion: I arrived the discussion waiting to read some statistical significance tests of association between your variables; I believe you need to improve your study with some more statistical content than percentages, there are so many variables to associate!

We have conducted statistical analysis using The Khi2 test to assess for significant differences between the calculated frequencies and proportions and associations were assessed using the chi-square test for independence, accompanied by a correlation coefficient (Phi or Cramer’s V), however the results were not conclusive that is why we have decided to omit it from the manuscript

-      Line 246 – AI in full (or in first report write it full plus (AI))

Done

-      You should add the limitations of your study and I believe you can enrich the discussion much more with a better statistical analysis.

We thank the author for this comment. Of course, the study has limitations, we have added a section with the study limitations

Reviewer 2 Report

Dear authors, 

Interesting work, good to identify the problemacy. 

I have some questions that came up throughout the manuscripts:

Do you see a link between some diseases in commercial poultry farms and diseases reported in backyard poultry? 

Did you check the birds/ night shelter during the questionnaires? 

Do you have a list of products used (veterinary medicines, antibiotics, other drugs, pesticides, other products used (eg. plant based extracts)?

Line 53: How is sustainability measured or evaluated (Sociological, economical, environmental)? 

Line 190: Off-farm disposal of birds: where, how? What about the effect on environment? 

Line 248: Do you have more information on worms or poultry red mites? 

Author Response

Response to Reviewer 2 Comments

Do you see a link between some diseases in commercial poultry farms and diseases reported in backyard poultry? 

We thank the reviewer for this comment. Certainly, from the clinical signs described by the flock owners, we have concluded that backyard poultry were affected by the same major infectious disease that occurs in commercial farms and we assumed that the backyard poultry can act as a reservoir of these pathogens.

Did you check the birds/ night shelter during the questionnaires? 

Yes, during the questionnaire, the night shelters and hygiene and biosecurity practices were audited and checked

Do you have a list of products used (veterinary medicines, antibiotics, other drugs, pesticides, other products used (eg. plant based extracts)?

We have kept a list of all the drug, chemicals and ethnomedical remedies used by the flock owners

For antibiotics, the most used was the sulfamethoxazole, trimethoprim and amoxicillin (prescribed for human use)

For pesticides, the flock owners purchase pesticides in retail. For example, the brand Nuvan (dichlorvos) and other smuggled brand such as “Elguenfoud”

Fr ethnomedical remedies, the flocks owners usually use onion, nettles, garlic, species and in some cases, they use the bleach to treat or to promote growth of their poultry

Line 53: How is sustainability measured or evaluated (Sociological, economical, environmental)? 

In that study, the sustainability was assessed sociologically and economically

Line 190: Off-farm disposal of birds: where, how? What about the effect on environment? 

The off-farm disposal of dead birds was practiced any where in the nature and some of the respondents stated that they throw carcasses even in the near streams and pounds without any concern about the potential risk and environmental impact of such practice

Line 248: Do you have more information on worms or poultry red mites? 

From the present survey, we have found that ectoparasite infestation mainly with red mites was the most common disease reported by flock owners especially during hot seasons. and that is why the Ectoparasite treatment was very common.

Round 2

Reviewer 1 Report

Dear authors,

In your review you limited yourself to correcting the English or formatting issues that I suggested, however you did not give concrete answers or improvements to more important issues such as discussion or statistical treatment. Concerning this last one, as said before, you should replace all the “,” by “.” in percentages but mostly, you claimed that the results were "inconclusive" - statistic is not never "inconclusive" - but you
added in all variables p<0.0001 (!!!) which is very "conclusive" and significant. And strange also that all the variables have a p<0.0001. You should address this issue.

Author Response

Response 1:

Dear reviewer,

First of all, I want to thank you for your valuable comments that allow me to improve my manuscript.

Concerning the discussion and statistical analysis, unfortunately, in the present study I could not find a lot of associations between the studied variables, However using the Khi square test, I have found some relevant associations :

  • The first one was between the practice of vaccination and the flock owner’s gender and level of education
  • The second association was found between veterinary care and the presence of a commercial poultry set nearby.
  • The last association was found between the contact of wild birds and confinement system and water source.

I added these associations and their interpretation and discussion to the manuscript, however, I don’t know if I should also add the table below to the manuscript or if I can simply mention that I found the associations in the manuscript since there are just a few meaningful associations.

Regarding the Khi square test of equality of proportions, several variables show unequal proportions and a p-value of <0.001 or <0.05, Only the cleaning of feeders and drinkers showed equality of proportions.

Table x. p values for Chi-square tests between studied variables

Variables

Location

Gender

Education level

Confinement system

Method of distribution

Vaccination

Presence of industrial poultry farms nearby

Contact with wild birds

Location 

-

<0,05

<0,001

<0,001

<0,05

NS

NS

NS

Gender

<0,05

-

<0,001

NS

<0,05

<0,001

NS

NS

Education level

<0,001

<0,001

-

NS

NS

<0,05

NS

NS

Confinement system

<0,001

NS

NS

-

NS

NS

NS

<0,05

Cleaning and disinfection of shelters

NS

NS

NS

<0,001

NS

NS

NS

NS

Supplementation

NS

NS

NS

NS

<0,001

NS

NS

NS

Water source

NS

NS

NS

NS

NS

NS

NS

<0,05

Vaccination

NS

<0,001

<0,05

NS

NS

-

NS

NS

Veterinary care

NS

NS

NS

NS

NS

NS

<0,001

NS
